# GRADIENT-NORM CONSTRAINED ALGORITHM ON OFFLINE AND ONLINE LEARNING

## ABSTRACT

Reinforcement learning (RL) has displayed great potential on both discrete and continuous tasks. However, its applicability in in realistic settings is curbed by the inherent uncertainty in value estimates and policy estimates, especially for continuous control problem. We propose an off-policy actor-critic method for deep reinforcement learning (DRL), where the value and policy are estimated by function approximation. The gradient decent approach to update parameters of multiple neural networks alternatively will inevitably cause the saddle point problem, which will degrade the overall convergence performance of training processes. Besides, off-policy methods can induce distribution mismatch, causing a deadly cycle of overestimation, when the candidate policies are conspicuously different from the policy which produces the data in replay buffer. Therefore, despite enjoying the advantage in sample complexity, the off-policy actor-critic methods is highly sensitive to network initialization, especially in the absence of expert demonstrations. We attempt to tackle these two issues by proposing a novel policy regularization and related value penalty, respectively. The policy regularization makes the training less content with the saddle point which pretends to be an optimal one and encourages the training to skip it. And the value penalty discourages over-optimistic value estimates. The proposed method is further combined with behavior cloning to apply to offline RL and tested on D4RL benchmarks.

## 1 INTRODUCTION

In recent years, DRL has demonstrated great potential in recent year. However, its application is mainly limited to video games Mnih et al. (2015); Silver et al. (2016; 2018) that can virtually interact with the related environment. In Markov Decision Process (MDP) settings, the agent takes an action on the basis of the current state in the environment and then observe the reward and next state. To efficiently exploit the experience data, we focus on the off-policy issues where the policy underlying the estimated value is evaluated by a dataset collected according to another behavior policy Sutton & Barto (2018). Off-policy methods usually use a replay buffer to store the states, actions and observations as the experience data, which is quite significant in real-world applications because the dataset of the replay buffer constitutes the heuristic basis of offline RL.

Although off-policy RL algorithms adopt mini-batches sampled from previously-collected dataset during the training process, it does not always perform well due to the extrapolation error induced by the mismatch between the distribution of experience dataset and the state-action visitation of the current policy Fujimoto et al. (2019). The main challenge encountered by related algorithms is that the coverage of policy visitation is not enough for further evaluations, which can cause the out-of-distribution (OOD) actions. The situation is even worse when OOD actions predict inaccurate value estimates that outweigh the true value Thrun & Schwartz (1993), i.e., the overestimation, since RL tends to choose actions pursuing value maximization. In practice, compared with underestimaton, overestimation is less preferable because it further accumulates and broadcasts via bootstrapping of temporal difference learning Sutton & Barto (2018).

In continuous control setting, the actor-critic approach is unavoidably adopted due to the flexibility brought by the function approximation. However, the induced intrinsic noise will aggravate the inaccuracy of value estimates and thus cause estimation errors. Besides, the saddle point problem will be exaggerated by the uncertainty of function approximation, and cause further suboptimal

policies and fake convergence. In general, the saddle point problem is extremely difficult to solve because of the nature of DRL, where the algorithm designer can only link the input actions and observations through a "black box". In comparison, the value overestimation is much easier to be addressed. There have been several categories of methods developed to tackle the overestimation caused by distribution shifts, including the policy constraint, V-function and conservative estimation Zhao et al. (2022). The V-function is distinct from the widely used actin-value (Q-value) function because it does not rely on the action and thus avoid optimizing the policy heuristically. The policy constraint explicitly constrains agent's policy into behavior policy or uses divergence to keep policy close to behavior policy. The conservative estimation reduces the value of OOD actions so that the in-distribution actions have higher probability to be selected. It mitigates the policy extrapolation errors while compromising with policy exploration. Therefore, the conservative estimation can be realized by value penalty, and assisted with proper policy regularization, which is partly the attempt of our work.

When we try to deal with standard off-policy RL, we need to mention the offline or batch RL, whose dataset is given by some expert demonstrations instead of real environment interactions. Compared with offline RL, online off-policy RL has the ability to test and adjust the policy trough environment interactions. However, the attempted exploration may be false or invalid due to inaccurate value estimates, which may result in the worse performance than the offline counterpart with good fixed dataset. Except for the sources of dataset, offline RL and online off-policy RL share the same problems in overestimation errors and saddle point. Therefore, we believe that our proposed approach to off-policy RL can also apply to offline RL.

The paper aims to address the value overestimation and saddle point problems to get a better policy appropriately while keeping the Q-values estimates relatively accurate. Specifically, to avoid suboptimal or fake convergence, we first make the worst assumption that every candidate of optimal point is suboptimal and make the agent get over it. This is realized by a gradient-based policy regularization. We propose that the gradient of Q-value with respect to the action should be bounded by a hard constraint. Second, to avoid the second order gradient of Q-value with respect to the action being zero, we add some nonlinearity into the fully-connected critic layers. Third, we propose a value penalty concerned with the norm of Q-value gradient with respect to the action to coordinate with the regularization term and relieve the negative impact of overestimation. After that, we present a novel algorithm which combines our method with a certain off-policy actor-critic approach to tackle the above mentioned problems and pursue effective and stable exploration. Furthermore, the proposed method is further constrained by behavior cloning to observe its offline performance, which is evaluated on D4RL benchmarks of MuJoCo tasks. Finally, experimental evaluations are conducted on a set of Gym tasks including HalfCheetah, Hopper, Walker2D, Ant and Humanoid, to compare the proposed algorithm with several baselines in terms of sample efficiency and stability.

## 2 RELATED WORK

Our work deals with the off-policy problems with the actor-critic method, which starts from policy iteration and alternates between policy evaluation and policy improvement steps, computing the value function and deriving an updated policy, respectively. In off-policy RL, the agent has to balance the tradeoff between exploration and exploitation. Without exploration, the policy is unable to improve by itself and the value evaluation cannot proceed to reach the fixed point for the overall performance of training. Given no expert dataset, the agent needs to interact with the environment to gain experience to optimize the RL objective. The exploration issue is addressed by maximum entropy learning in the up-to-date literature Ziebart et al. (2008); Toussaint (2009); Rawlik et al. (2012); Fox et al. (2015); Haarnoja et al. (2017), which can substantially improve the robustness when dealing with uncertainty errors Ziebart (2010). Maximum entropy learning improves exploration by broaden the diversity of behaviors Haarnoja et al. (2017), and its combination with the actor-critic architecture, which separates the policy network from the value function, encourages the action exploration and makes some efforts to combat the suboptimal convergence Barto et al. (1983); Sutton & Barto (2018).

Policy entropy is also considered to belong to the family of regularizer Schulman et al. (2015; 2017); Mnih et al. (2016); Gruslys et al. (2017), which is usually adopted to cope with erroneous action-value estimates caused by function approximation and distribution shifts Sutton (1995); Baird

(1995); Tsitsiklis & Van Roy (1996); Van Hasselt et al. (2018). The solutions to these errors can be divided into two directions, i.e., the value penalty and policy regularization. Based on the intuition that the negative effect of overestimation is more destructive on OOD actions, the policy regularization or constraint is often realized by constraining agent's policy into the behavior policy or regularizing the learned policy towards the behavior policy through some divergence criterion Fujimoto et al. (2019); Kumar et al. (2019); Jaques et al. (2019); Laroche et al. (2019). The maximum entropy policy regularization Haarnoja et al. (2018a;b) adopts stochastic policies to generalize the policy improvement and introduce uncertainty into action decisions over deterministic counterparts Heess et al. (2015). A typical approach to the value penalty is sophisticated ensembles of target Q-value to remedy the value overestimation and stabilize the trained Q-value function Fujimoto et al. (2019); Kumar et al. (2019); Agarwal et al. (2019). Another way to avoids overestimation induced by OOD actions is the conservative estimation, which assigns lower value to potential state-action pairs Kumar et al. (2020); Kostrikov et al. (2021). Specially, Zhao et al. (2022) changes the degree of conservatism in training by adding a simple constant term to the estimated Q-value, and Fakoor et al. (2021) proposes a value-constraint to discourage over-optimistic value estimates and a policy constraint to reduce the divergence between the learnt policy and the unknown behavior policy, respectively, involving several divergence metrics Wu et al. (2019).

## 3 Preliminaries

In this work, we focus on the Markov Decision Process (MDP). A standard MDP can be represented by the tuple $(\mathcal{S}, \mathcal{A}, T, r)$ where $\mathcal{S}$ is the state space, $\mathcal{A}$ is the action space, $T(\cdot|s, a)$ is the transition probability of the next state $s' \in \mathcal{S}$ conditioned on the current state $s \in \mathcal{S}$ and action $a \in \mathcal{A}$, and $r \in \mathcal{S} \times \mathcal{A}$ is the reward which is the feedback from the environment of the current state $s$ and action $a$. The objective of RL is to find a policy to maximize the expected return denoted by the discounted cumulative reward. When the function approximation is adopted in actor-critic RL, both the policy and value function are parameterized by neural networks. Let $\pi$ denote the policy, then the objective of expected return with respect to the state-action pair can be given in the form of

$$Q_\pi(s, a) = \mathbb{E}_{p^\pi(s_t|s_0, a_0)} \left[ \sum_{t=0}^{\infty} \gamma^t r(s_t, a_t) | s_0 = s, a_0 = a \right], \tag{1}$$

where $r(s, a)$ is the immediate reward produced by the state-action pair, and $\gamma \in (0, 1)$ is the discount horizon factor for future rewards. With the effect of behavior policy $\pi$, $p^\pi(s_t|s_0, a_0) = T(s_1|s_0, a_0) \prod_t \left[ \mathbb{E}_{a_{t-1} \sim \pi} T(s_t|s_{t-1}, a_{t-1}) \right]$ is the trajectory distribution of an episode given the initial state-action pair $(s_0, a_0)$, and $\pi(a_{t+1}|s_{t+1})$ indicates the conditional probability density function (pdf) for the agent to choose the action $a_{t+1}$ given the state $s_{t+1}$. We denote $a = \pi(s)$ as the generation function mapping the state to the action, which is distinct from the conditional pdf $\pi(a|s)$.

### 3.1 Saddle Point Problem

The structured saddle point often shows up in the form of optimizing an objective over multiple variables, which is applied in generative adversarial networks, robust optimization Ben-Tal et al. (2009) and game theory Leyton-Brown & Shoham (2008); Singh et al. (2000). In actor-critic setting, the saddle point problem is produced by alternatively solving the optimization of value function and policy. To better illustrate the issue, we simply formulate our saddle point problem as

$$\min_\omega \max_\theta \mathbb{E}_{(s, a, r, s')} \left[ L(\omega) + J(\theta) \right], \tag{2}$$

where $\omega$ and $\theta$ are respectively the critic and actor parameters, $J(\theta) = \mathbb{E}_s \left[ Q_\omega(s, \pi_\theta(s)) \right]$, and

$$L(\omega) = (r + \gamma Q_{\omega'}^t(s', \pi_{\theta'}(s')) - Q_\omega(s, a))^2, \tag{3}$$

where $(s, a, r, s')$ is a transition slot sampled from the dataset at every environment step, and $\omega'$ and $\theta'$ are the parameters of target value and target policy which are correlated with $\omega$ and $\theta$.

Let $K(\omega, \theta)$ denote the summation within the min-max operator in (2), then solving (2) is equivalent to find a point $(\omega^\star, \theta^\star)$ which satisfies the following condition that

$$K(\omega^\star, \theta) \leq K(\omega^\star, \theta^\star) \leq K(\omega, \theta^\star). \tag{4}$$

Assume the gradient of $K$ is Lipschitz with respect to the parameters $(\omega, \theta)$ First-order gradient descent or ascent is usually adopted to address RL, which gives the following iterative updates as

$$(\omega_{t+1}, \theta_{t+1}) = (\omega_t, \theta_t) + l_t(-\nabla_\omega L_t + \nabla_\theta J_t), \tag{5}$$

where $l_t$ is the learning rate, which is often selected to be time-decreased or a constant. Solving (4) can reduce to a locally optimal saddle point $(\omega^\triangle, \theta^\triangle)$, which meets $\nabla K(\omega^\triangle, \theta^\triangle) = 0$. Considering the usage of updating rule in (5), the property of zero gradient makes saddle point a candidate of converged points. In another words, the gradient-based optimization is apt to converge to some undesired stable stationary points, which are not able to escape through successive iterations of gradient descent or ascent Adolphs et al. (2019).

## 4 METHODS

Our previous discussion on the problems existing in off-policy actor-critic RL suggests two aspects of fixes that have the potential to improve the convergence while arriving at more precise Q-values estimates. To address these, we first have to restrict the value overestimation through value penalty, which is modeled as the norm of the partial gradient of Q-value function with respect to the action. The heuristic intuition underlying this method is to penalize the value estimation at regions with high sharpness of Q-values with respect to actions. Secondly, we enable the agent to escape from the gravitation of any saddle point so that the final converged value can grow naturally with fewer unreasonable caps. To achieve this goal, we have to make a conservative assumption that all stationary points encountered by the agent during the training are saddle points. We further optimize the policy with a constraint set upon the partial gradient of Q-value with respect to the action. The commonly used Multi-Layer Perceptron (MLP) will reduce the partial gradient of Q-value with respect to the action to be constant, so we reorganize the layer design.

### 4.1 GRADIENT-NORM CONSTRAINED VALUE ITERATION

The iteration of gradient-norm constrained (GNC) value is started by determining the target value for the critic based on TD learning and Bellman equation, which is the selection of $Q_{\omega'}^t$ as shown in (3). In standard deep deterministic policy gradient (DDPG) Lillicrap et al. (2015), $Q_{\omega'}^t$ is exactly the same as $Q_{\omega'}$. When the entropy term is adopted in value penalty, like soft actor-critic (SAC) Haarnoja et al. (2018a;b), the target value is penalized by an entropy term to make the Q-value estimate comply with the maximum entropy policy. Though in different forms, the entropy-based value penalty seems to belong to the family of conservative estimation, which gives lower value to selected regions of state-action pairs that have the potential to cause policy extrapolation errors. Recently it is reported in Zhao et al. (2022) that a constant subtracted from $Q_{\omega'}$ can effectively mitigate the overestimation of OOD actions. Overall, the conservative estimate of target Q-value has the form of $Q_{\omega'}^t = Q_{\omega'} - penalty$, where the $penalty$ is usually nonnegative.

In our work, we model the value penalty as the second-order norm of the partial gradient of Q-value function with respect to the action to approximate the sharpness of Q-value. Given the MDP denoted by $(\mathcal{S}, \mathcal{A}, p, r)$, then a modified Bellman backup operator $\mathcal{T}^\pi$ is given by

$$\mathcal{T}^\pi Q(s_t, a_t) = r(s_t, a_t) + \gamma \mathbb{E}_{s_{t+1}, a_{t+1}} \left[ Q^t(s_{t+1}, a_{t+1}) \right], \tag{6}$$

where $s_{t+1} \sim T(\cdot|s_t, a_t)$ and $a_{t+1} \sim \pi(\cdot|s_{t+1})$, and

$$Q^t(s_{t+1}, a_{t+1}) = Q(s_{t+1}, a_{t+1}) - \beta \|\nabla_{a_{t+1}} Q(s_{t+1}, a_{t+1})\|_2, \tag{7}$$

is the gradient-norm constrained value function, and $\beta$ is a fixed newly-induced hyperparameter, which is used to balance the contribution of GNC term to avoid over-punishment. By this means, the agent are capable of penalizing the unstable gradients since the misleadingly sharp partial Q-value gradients is inclined to get into OOD actions. In prior methods, the value penalty term is often unrelated with the Q-value function, so that their convergence will not be affected by the extra term. Although our value penalty is correlated with the partial gradient of Q-value, we can still prove that repeatedly employing $\mathcal{T}^\pi$ for any policy $\pi$ will end up with a Q-value floating around the fixed point of Bellman operator.

**Lemma 1.** *Given the condition that $\nabla_a Q(s, a)$ is well defined, and $\exists L_r, L_Q, L_\nabla > 0$ such that $|r(s, a)| \leq L_r, |Q(s, a)| \leq L_Q, \|\nabla_a Q(s, a)\|_2 \leq L_\nabla, \forall (s, a) \in \mathcal{S} \times \mathcal{A}$, then the sequence $Q_{w_{k+1}}(s_t, a_t) = \mathcal{T}^\pi Q_{w_k}(s_t, a_t)$ will be bounded around a fixed point as $t \to \infty$.*

The proof of Lemma 3 can be found in Appendix A. According to the proof, how far it deviates from the original fixed point can be controlled by the hyperparameter of $\beta$. Throughout this proof, we did not use the original definition of Q-value in (1), since the function approximation is often adopted in DRL to replace (1). However, (1) constitutes the basis of Bellman operator. The requirement for the 2-norm of the partial gradient of Q-value function w.r.t. the action to be bounded is weak and can be satisfied by rearranging the critic network. In comparison to the work in Gao et al. (2022), which proposes a sufficient condition that the norm of the partial gradient of policy $\pi$ w.r.t. the action should be bounded below unit to ensure that the norm of the partial gradient of Q-value function w.r.t. the action to be bounded, our condition is looser because the design of actor network is subjected to normalization except for the hard constraint less than 1. Since the state-action spaces are continuous and the transition probability is unknown in model-free DRL, the Q-value function cannot be formulated or tabulated by the state-action pairs, which means the function approximation gives no absolute guarantee for the bounded Q-values. Therefore, instead of repeatedly applying (6) directly, the practical evaluation step is estimated by minimizing the expected mean square error (MSE) between $\mathcal{T}^\pi Q(s, a)$ and $Q(s, a)$. Once the expected MSE converges to be acceptably small, the updates of Q-value function based on MSE will end up with little fluctuation around the fixed point when the $\beta$ is proper.

## 4.2 GRADIENT-NORM CONSTRAINED POLICY REGULARIZATION

When it comes to the policy regularization, we try to avoid any candidate of saddle points. However, due to the inaccurate value estimates and unknown transition probability, the distinction between the optimal point and saddle points is invisible to algorithm developers. Therefore, we treat all stationary points to be saddle points to make a compromise between the asymptotic performance and convergence rate. Based on this idea, we optimize the policy subjected to a condition that the partial gradient of Q-value w.r.t. the action should be nonnegative. We give the constrained policy regularization as

$$\max_\theta \mathbb{E}_s \left[ Q_\omega(s, \pi_\theta(s)) \right] \quad s.t. \quad \mathbb{E}_s \left[ \nabla_\theta Q_\omega(s, \pi_\theta(s)) \right] \geq 0, \tag{8}$$

where $\pi_\theta$ is the policy approximation parameterized by $\theta$, which is often modeled as the standard deviation of a Gaussian distribution concerned with the action. The heuristic intuition underlying this constraint is to encourage the agent to skip the current stationary point and find the next one. However, this constraint is not easy to address in this form, we change it into some dual form instead.

**Proposition 1.** *Given the condition that the policy is parameterized as a Gaussian distribution w.r.t. the action, then the following condition that*

$$\mathbb{E}_s \left[ \sum_{i=0}^{|\mathcal{A}|} \left( \nabla_{a_i} Q_\omega(s, a_i) \right) \right] \geq 0, \tag{9}$$

*where $|\mathcal{A}|$ represents the dimension of action space and $a_i$ is the $i$-th element of the action vector, is sufficient for $\mathbb{E}_s \left[ \nabla_\theta Q_\omega(s, \pi_\theta(s)) \right] \geq 0$.*

The proof of Proposition 2 can be found in Appendix B. We use the summation of all elements of the Q-value's partial gradient w.r.t. the action to weight its contribution because $\nabla_a Q_\omega(s, a)$ is a vector. Since the constraint optimization problem can be solved by Lagrange dual form, we can project the policy onto a unconstrained normalized distribution, which is given by

$$a = \pi_{new}(s) = \arg\min_\pi D_{KL} \left( \pi(\cdot|s) \| e^{\frac{Q^\pi(s,\cdot) - \frac{\beta}{\sqrt{|\mathcal{A}|}} \sum_\cdot \nabla_\cdot Q(s,\cdot)}{\alpha}} \right)$$

$$= \arg\max_\pi \mathbb{E}_a \left[ Q(s, a) - \frac{\beta}{\sqrt{|\mathcal{A}|}} \sum_{i=0}^{|\mathcal{A}|} \nabla_{a_i} Q(s, a_i) \right], \tag{10}$$

where $s \sim \mathcal{S}, a \sim \pi(\cdot|s_t)$, $D_{KL}(\cdot \| \cdot)$ is the KL divergence, $Q^\pi(s, \cdot) = Q(s, \cdot) + \alpha \log(\pi(\cdot|s))$ is the revised value to resist the extra exploration weighted by the temperature parameter $\alpha$, and the choice of policy $\pi$ is limited to a set of parameterized Gaussian distributions for flexibility. The equality in (22) holds with modest computation which can be found in Appendix C. The KL divergence $D_{KL}$

shows that the improved policy is updated towards the distribution constituted by the exponential of the normalized Q-value function.

In model-free RL problems on continuous control with function approximation, where the transition probability is unknown in continuous state and action spaces, it is extremely difficult to provides pointwise policy improvement (absolute policy improvement) over $\mathcal{S} \times \mathcal{A}$. Therefore, we propose a looser but more practical criterion for the policy improvement, which is denoted as the expected policy improvement in this paper. In general, maximizing the Q-value or keeping the policy regularization in the same form with the value penalty will simply produce higher updated Q-value formulated by (1). We show that the projected policy in (22) coordinated with the value update as (6) remains the virtue of policy improvement, and the result is organized in Lemma 4.

**Lemma 2.** *Denote $\pi_{new}$ and $\pi_{old}$ as the policies before and after the update defined in* (22)*, respectively. Then the expected policy improvement, i.e., $\mathbb{E}_{(s_t,a_t) \sim \mathcal{S} \times \mathcal{A}}[Q_{\pi_{new}}(s_t, a_t) - Q_{\pi_{old}}(s_t, a_t)] \geq 0$, can be guaranteed, where $Q_{\pi_{new}}$ is the real Q-value following bellman operator and $Q_{\pi_{old}}$ is the modified value employing* (6)*.*

The proof of Lemma 4 can be found in Appendix D. Besides projecting the policy into a selected set of distributions, (22) also normalizes the regularization term by $|\mathcal{A}|$, which is the key to the guarantee of the wanted expected policy improvement, shown in the last step of proving Lemma 4. In discrete control problems, where the state-action spaces are both discrete and bounded, the pointwise policy improvement may be realized without the expectation over $(s, a) \sim \mathcal{S} \times \mathcal{A}$ in Lemma 4.

The GNC iteration alternates between the policy evaluation and the expected policy improvement steps, and can be summarized by combining Lemma 3 and Lemma 4 as following

**Theorem 1.** *Given the condition that $\nabla_a Q(s, a)$ is well defined, and all the reward function, approximated Q-value function and its partial gradient w.r.t. the action are bounded throughout the state-action space, then repeatedly employing* (6) *will finally reach a value bounded around a fixed point. If the policy updating rule follows* (22)*, the real Q-value following bellman operator after updating is no less than the modified value employing* (6) *before updating in expectation form.*

### 4.3 GRADIENT-NORM CONSTRAINED ALGORITHM

We have discussed above conditions of Theorem 1 in large continuous domains, which requires parameterized function approximations to estimate both the Q-value and the policy. Prior works in the literature usually adopt a separated target network to stabilize the training process for both the Q-value and the policy. Given the current and target networks of both the Q-value and the policy and on the basis of (6), the loss function for the update of critic parameters in the policy evaluation step can be estimated by

$$L(\omega) = \mathbb{E}_{(s,a,r,s')} \left[ \frac{1}{2}(r + \gamma Q_{\omega'}^t(s', a') - Q_\omega(s, a))^2 \right], \tag{11}$$

$$Q_{\omega'}^t(s', a') = Q_{\omega'}(s', a') - \beta \|\nabla_{a'} Q_{\omega'}(s', a')\|_2, \tag{12}$$

where $a' = \pi_{\theta'}(s')$ is the action following the target policy parameterized by $\theta'$, $(s, a, r, s')$ is a tuple of history data sampled from the experience pool, $\omega$ and $\omega'$ parameterize the critic network and its target estimate, respectively, and $\pi_{\theta'}(\cdot|s')$ is the target policy pdf conditioned on the next state $s'$.

By minimizing (11), the critic parameters can be updated at every policy evaluation step with stochastic gradient as

$$\nabla_\omega L(\omega) = \mathbb{E}_{(s,a,r,s')}[\nabla_\omega Q_\omega(s, a)(Q_\omega(s, a) - r - \gamma Q_{\omega'}^t(s', a'))]. \tag{13}$$

It is noticeable that no additional network is induced in GNC value penalty or policy regularization, only the hyperparameter $\beta$ is adopted to control the contribution of GNC related terms. Then the surrogate objective function to update the current actor parameter $\theta$ in the expected policy improvement step (see Lemma 4) can be given by

$$J(\theta) = \mathbb{E}_s \left[ Q_\omega(s, \pi_\theta(s)) - \frac{\beta}{\sqrt{|\mathcal{A}|}} \sum_{i=0}^{|\mathcal{A}|} \nabla_{a_i} Q_\omega(s, a_i) \right], \tag{14}$$

where $a = \pi_\theta(s)$ is the reparameterized action with parameter $\theta$ based on $s$ sampled from the tuple of history data, and $a_i$ is the $i$-th element of the action vector. By maximizing (14), the actor parameter can be updated at every policy improvement step. The gradient of (14) is computed as

$$\nabla_\theta J(\theta) = \mathbb{E}_s \left[ \nabla_a Q_\omega(s, a) - \frac{\beta}{\sqrt{|\mathcal{A}|}} \sum_{i=0}^{|\mathcal{A}|} \nabla_{a_i}^2 Q_\omega(s, a_i) \right] \cdot \nabla_\theta \pi_\theta(s), \tag{15}$$

where $\nabla_{a_i}^2$ represents the second order partial gradient w.r.t. $a_i$.

---

**Algorithm 1** GNC Algorithm

---

1: Initialize parameters $\omega \leftarrow \omega_0, \theta \leftarrow \theta_0$
2: Initialize target parameters $\omega' \leftarrow \omega'_0, \theta' \leftarrow \theta'_0$
3: Initialize the learning rates $l_c, l_a$ for the critic and the actor, the time step $t \leftarrow 0$, the soft update hyperparameter $\tau$, the maximum time step $T$, the batch size $B$ and the replay buffer $\mathcal{D} \leftarrow \emptyset$.
4: **while** $t < T$ **do**
5:     Select action $a_t \sim \pi_{\theta_t}(a_t|s_t)$
6:     Observe the reward and next state $s_{t+1}, r_t \sim T(s_{t+1}|s_t, a_t)$
7:     Store transition $\mathcal{D} \leftarrow \mathcal{D} \cup \{(s_t, a_t, r_t, s_{t+1})\}$
8:     Sample a batch of transitions $\mathcal{B} = (s, a, r, s')_{i=1}^B$ from $\mathcal{D}$
9:     **for** each time step **do**
10:         $\omega_{t+1} \leftarrow \omega_t - l_c \nabla_{\omega_t} L(\omega_t)$ following (13)
11:         $\theta_{t+1} \leftarrow \theta_t + l_a \nabla_{\theta_t} J(\theta_t)$ following (15)
12:         $\omega'_{t+1} \leftarrow \tau \omega_{t+1} + (1 - \tau)\omega'_t$
13:     **end for**
14:     $s_{t+1} \leftarrow s_t$
15:     $t \leftarrow t + 1$
16: **end while**

---

### 4.4 GRADIENT-NORM CONSTRAINED ALGORITHM WITH BEHAVIOR CLONING

The extrapolation error commonly happens to offline RL and can be can be interpreted as the incompetence to address OOD actions. Considering this issue, priors approaches resort to varieties of regularizing methods to make the OOD actions easier to be identified. Although there are works trying to minimize some distance metrics, like the KL divergence, Fujimoto & Gu (2021) simply adopts minimal modifications to pre-existing twin delayed deep deterministic policy gradient (TD3) Fujimoto et al. (2018) to reduce the extrapolation error. By applying the minimal modifications, some benefits can be achieved, for example, not adding the variable parameters, reducing the computational complexity and providing analysis avenue for future methods. The algorithm proposed by Fujimoto & Gu (2021) is built on top of TD3 and named as TD3+BC, which makes a compromise between TD3 and behavior cloning (BC). There are two changes of TD3+BC, the first one of which is adding a behavior cloning regularization term to the policy improvement step of TD3 so that the policy can be pushed towards actions contained in the dataset. The second change is normalizing the Q-value term to strike a balance between it and the BC term. Overall, the method of combining the actor update term and the BC term with a hyperparameter to tune their relative weight can be applied to our algorithm.

When applying similar modifications to our algorithm, the two main changes are the dataset and the policy improvement step. In offline RL, the dataset is collected by interacting with the environment, while the dataset of online RL is given by prior demonstrations. Therefore, a successful policy has to be explored around the dataset, in other words, the policy far away from the distribution of given dataset will generate failing OOD actions. In summary, we balance the policy exploration and imitation as following

$$\theta = \arg\max_\theta \mathbb{E}_{(s,a)\sim\mathcal{D}'} \left\{ \lambda \left[ Q_\omega(s, \pi_\theta(s)) - \frac{\beta}{\sqrt{|\mathcal{A}|}} \sum_{i=0}^{|\mathcal{A}|} \nabla_{a_i} Q_\omega(s, a_i) \right] - (\pi_\theta(s) - a)^2 \right\}, \tag{16}$$

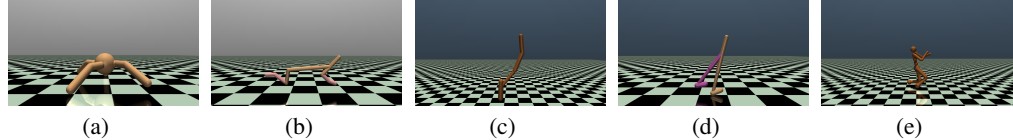

Figure 1: (a) Ant; (b) Halfcheetah; (c) Hopper; (d) Walker2d; (e) Humanoid

where $\mathcal{D}'$ represents the given dataset, and $\lambda$ is a hyperparameter that balances the values of original actor update term and the BC term. Since the scale of action is often assumed to be bounded, the balance can be easily broken by the Q-value that is highly susceptible to the scale of reward without any adjustment. In this case, $\lambda$ includes a normalization term to limit the contribution of Q-value. In our work, the normalization term is given by

$$\lambda = \frac{\xi}{\mathbb{E}_{(s,a)\sim\mathcal{D}'}\left[|Q_\omega(s,a)|\right]}, \tag{17}$$

where $\xi$ is a constant hyperparameter, and $(s, a)$ represents mini-batches sampled from $\mathcal{D}'$. The arrangement of $\lambda$ given by (17) is a convenient reference to Fujimoto & Gu (2021). However, the actor exploration term in (16) is different, then the GNC regularization term that follows the Q-value term will not degrade as the increase of Q-value's absolute value. By the way, the actor exploration term (actor update term) consists of the Q-value term and the GNC regularization term. Applying these modifications, we organize the GNC algorithm constrained by BC cloning as GNC-BC.

## 5 EXPERIMENTS

### 5.1 ONLINE BENCHMARKS AND BASELINES

Fig. 1 shows the illustrations of benchmarks adopted in this paper. The adopted baselines include DDPG, TD3 and SAC. Before the existence of SAC, DDPG is regarded as one of the most efficient off-policy DRL methods Duan et al. (2016), followed by TD3 as an extension. SAC has achieved model-free state-of-the-art sample efficiency in multiple challenging continuous control domains Christodoulou (2019).

Our proposed algorithm shares the same set of hyperparameters with other baselines to keep fairness. We organize the network architectures and hyperparameters in Appendix E and F, respectively. The Adam optimizer Kingma & Ba (2014) is used to update the network parameters. To keep a fair comparison, we train 5 seeds for each algorithm and plot the average reward versus the time step with recorded point at every 500 iterations (time steps). The results of algorithms are tested on selected benchmarks and shown in Fig. 2.

### 5.2 ONLINE RESULTS

For figures from Fig. 2(a)-2(e), we can see that GNC is more stable than all the selected baselines, which is revealed by the fact that the confidence interval (CI) of GNC is far less than that of other counterparts. The smallest CI is due to the low over-optimistic value estimates contributed by the proposed GNC value penalty in (12). Another noticeable observation from Figs. 2(a)-2(e) is that the converged value of GNC is the highest among all the baselines, especially in the Humanoid task. This phenomenon means GNC has stronger ability to break the cap set by undesired stationary points or saddle points, considering the contribution from the proposed GNC policy regularization in (14). We can also see that Fig. 2(e) shows overwhelmed advantage of GNC over other algorithms in both CI and converged value. We found the nonlinear inputs of GNC critic networks in Appendix E are not suitable for the Humanoid task, so $a^3$ is removed from the inputs of critic networks in the Humanoid task, which can be seen as a special case of our method. In this case, the partial gradient of Q-value w.r.t. the action is equal to the network parameters, which is similar to the constant penalty in Zhao et al. (2022). By heuristic intuition, conservative value estimate will result in lower convergence. However, the coordination between GNC policy regularization and value penalty leads to stable value improvement until convergence, as told in Lemma 4.

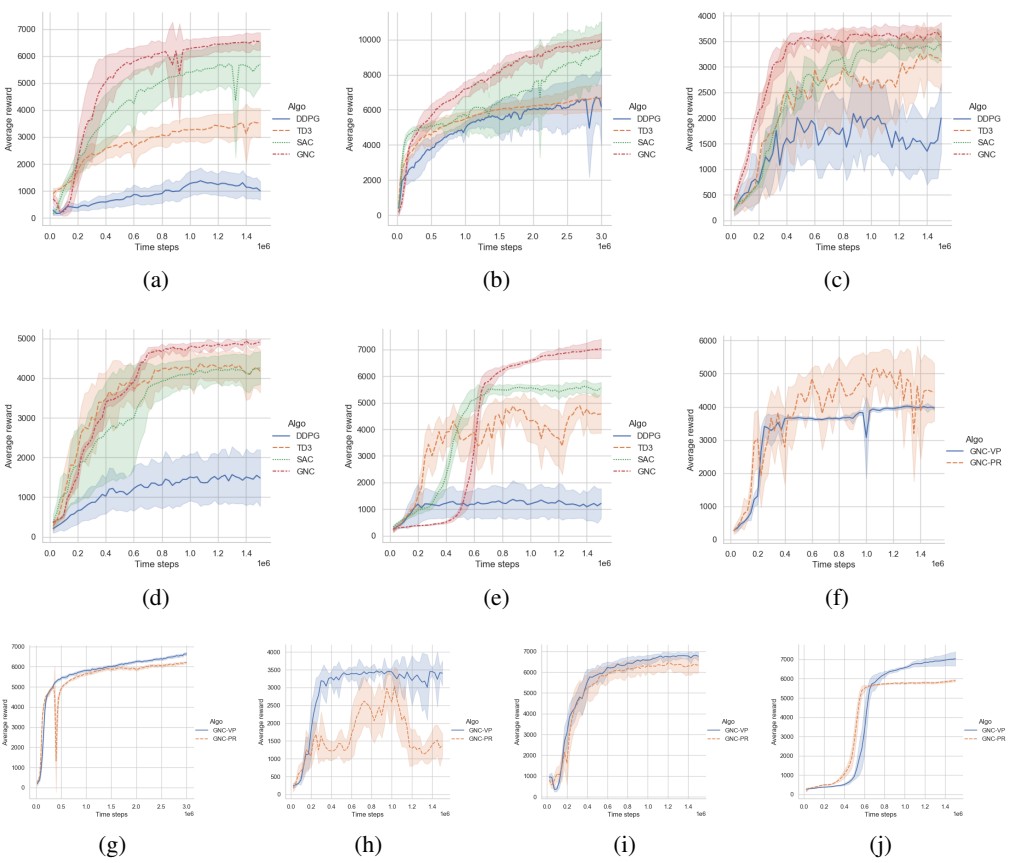

Figure 2: Average reward versus time step in (a) Ant; (b) Halfcheetah; (c) Hopper; (d) Walker2d; (e) Humanoid; (f) Walker2d ablation; (g) Halfcheetah ablation; (h) Hopper ablation; (i) Ant ablation; (j) Humanoid ablation

To better understand the two parts of GNC, we perform ablation studies by disabling either part. The results are illustrated in Figs.2(i)-2(j), in which GNC-VP and GNC-PR represents a variant GNC algorithm without the policy regularization and another variant without the value penalty term, respectively. It can be observed that GNC-VP or GNC-PR does not always perform well on these tasks, however, GNC-PR in Fig. 2(f) breaks 5000 score in Walker-2d during training, although it is unstable. In humanoid, GNC-VP is the same as GNC since $\nabla_a^2 Q_\omega$ in (15) is zero as mentioned above.

## 5.3 OFFLINE BENCHMARKS AND RESULTS

This part can be found in Appendix G.

## 6 CONCLUSION

In this paper, we propose a method based on a novel policy regularization and related value penalty, which attempt to tackle the saddle point problem and discourage the over-optimistic value estimates, respectively. Moreover, the proposed method is further constrained by behavior cloning to observe its offline performance and tested on D4RL benchmarks.

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

## A    PROOF OF LEMMA 3

**Lemma 3.** *Given the condition that $\nabla_a Q(s,a)$ is well defined, and $\exists L_r, L_Q, L_\nabla > 0$ such that $|r(s,a)| \leq L_r, |Q(s,a)| \leq L_Q, \|\nabla_a Q(s,a)\|_2 \leq L_\nabla, \forall (s,a) \in \mathcal{S} \times \mathcal{A}$, then the sequence $Q_{w_{k+1}}(s_t, a_t) = \mathcal{T}^\pi Q_{w_k}(s_t, a_t)$ will be bounded around a fixed point as $t \to \infty$.*

*Proof*

$$
\begin{aligned}
&\mathcal{T}^\pi Q(s_t, a_t) \\
=&r(s_t, a_t) + \gamma \mathbb{E}_{s_{t+1},a_{t+1}} \left[ Q(s_{t+1}, a_{t+1}) - \beta \|\nabla_{a_{t+1}} Q(s_{t+1}, a_{t+1})\|_2 \right] \\
\geq&r(s_t, a_t) - \gamma\beta L_\nabla + \gamma \mathbb{E}_{s_{t+1},a_{t+1}} \left[ Q(s_{t+1}, a_{t+1}) \right] \\
=&r(s_t, a_t) - \gamma\beta L_\nabla + \gamma \mathbb{E}_{s_{t+1},a_{t+1}} \left[ r(s_{t+1}, a_{t+1}) + \gamma \mathbb{E}_{s_{t+2},a_{t+2}} \left[ Q(s_{t+2}, a_{t+2}) \right] \right] \\
=&r(s_t, a_t) + \sum_{k=1}^{\infty} \gamma^k \mathbb{E}_{s_{t+k},a_{t+k}} \left[ r(s_{t+k}, a_{t+k}) \right] - \gamma\beta L_\nabla \\
=&r(s_t, a_t) + \frac{\gamma}{1-\gamma} \mathbb{E}_{s,a} \left[ r(s,a) \right] - \gamma\beta L_\nabla, \quad\quad\quad\quad (18)
\end{aligned}
$$

where the second equality is due to the Q-value function's Bellman expansion, and the last second equality holds because the Q-values are assumed to be bounded and $0 \leq \gamma < 1$ by iteratively employing the Bellman equation. By the way, $\mathbb{E}_{s_{t+k},a_{t+k}} \left[ r(s_{t+k}, a_{t+k}) \right] = \mathbb{E}_{s,a} \left[ r(s,a) \right]$ because of the expectation form. Similarly, we can also prove

$$
\mathcal{T}^\pi Q(s_t, a_t) \leq r(s_t, a_t) + \frac{\gamma}{1-\gamma} \mathbb{E}_{s,a} \left[ r(s,a) \right], \quad\quad\quad\quad (19)
$$

then by denoting $Q^\star = r(s_t, a_t) + \frac{\gamma}{1-\gamma} \mathbb{E}_{s,a} \left[ r(s,a) \right]$ as the fixed point, we can reach a conclusion that $Q^\star - \gamma\beta L_\nabla \leq \mathcal{T}^\pi Q(s_t, a_t) \leq Q^\star$. The fluctuation can be adapted by change $\beta$. $\quad\square$

# B  PROOF OF PROPOSITION 2

**Proposition 2.** *Given the condition that the policy is parameterized as a Gaussian distribution w.r.t. the action, then the following condition that*

$$\mathbb{E}_s \left[ \sum_{i=0}^{|\mathcal{A}|} (\nabla_{a_i} Q_\omega(s, a_i)) \right] \geq 0, \tag{20}$$

*where $|\mathcal{A}|$ represents the dimension of action space, is sufficient for $\mathbb{E}_s \left[ \nabla_\theta Q_\omega(s, \pi_\theta(s)) \right] \geq 0$.*

*Proof*  As mentioned, $a = \pi(s)$ is the generation function mapping the state to the action, and $\pi(a|s)$ is the conditional pdf of the action given the state, then $\pi(s) = F^{-1}(u|s)$, where $u$ is the uniform distribution ranging from $0$ to $1$, and $F^{-1}$ represents the inverse function of cumulative distribution of $\pi(a|s)$, then

$$
\begin{aligned}
&\mathbb{E}_s \left[ \nabla_\theta Q_\omega(s, \pi_\theta(s)) \right] \\
=&\mathbb{E}_s \left[ \nabla_a Q_\omega(s, a) \times \nabla_\theta a \right] \\
=&\mathbb{E}_s \left[ \nabla_a Q_\omega(s, a) \times (\nabla_\theta \pi_\theta(a|s))^{-1} \right] \\
=&\mathbb{E}_s \left[ \nabla_a Q_\omega(s, a) \times (2\overrightarrow{1_{|\mathcal{A}|\times 1}} \times \pi_\theta(a|s) \circ \theta^{-3})^{-1} \right] \\
=&2\mathbb{E}_s \left[ \nabla_a Q_\omega(s, a) \times \pi_\theta^{-1}(a|s) \circ \theta^3 \right] \\
\geq&2 \arg\min_{\overline{a}} \mathbb{E}_s \left[ \nabla_a Q_\omega(s, a) \times (\pi_\theta^{-1}(\overline{a}|s)\overrightarrow{1_{|\mathcal{A}|\times|\theta|}}) \circ \theta^3 \right] \\
=&2\pi_\theta^{-1}(\overline{a}|s)\mathbb{E}_s \left[ \nabla_a Q_\omega(s, a) \times \overrightarrow{1_{|\mathcal{A}|\times|\theta|}} \circ \theta^3 \right] \\
=&2\pi_\theta^{-1}(\overline{a}|s)\mathbb{E}_s \left[ \sum_{i=0}^{|\mathcal{A}|} (\nabla_{a_i} Q_\omega(s, a_i)) \overrightarrow{1_{1\times|\theta|}} \circ \theta^3 \right] \\
=&2\pi_\theta^{-1}(\overline{a}|s)\mathbb{E}_s \left[ \sum_{i=0}^{|\mathcal{A}|} (\nabla_{a_i} Q_\omega(s, a_i)) \right] \overrightarrow{1_{1\times|\theta|}} \circ \theta^3, \tag{21}
\end{aligned}
$$

where we simply use $(\cdot)^{-1}$ and $\circ$ to stand for the elementwise reciprocal and elementwise product, respectively, and $\overrightarrow{1_{m\times n}}$ means a matrix of $m \times n$ with all ones. Since the actions are bound, there exists a scalar $\overline{a}$ to minimize $\mathbb{E}_s \left[ \nabla_a Q_\omega(s, a) \times \pi_\theta^{-1}(a|s) \right]$, then the inequality of above derivation holds, and $\pi_\theta^{-1}(\overline{a}|s)$ is also a scalar. Since both the conditional pdf and $\theta$ are nonnegative, $\mathbb{E}_s \left[ \sum_{i=0}^{|\mathcal{A}|} (\nabla_{a_i} Q_\omega(s, a_i)) \right] \geq 0$ can ensure (21) to be nonnegative.

$\square$

# C  PROOF OF (22)

$$
\begin{aligned}
a &= \pi_{new}(s) \\
&= \arg\min_{\pi \in \prod} D_{KL} \left( \pi(\cdot|s) \| \exp \left( \frac{Q^\pi(s, \cdot) - \frac{\beta}{\sqrt{|\mathcal{A}|}} \sum. \nabla. Q(s, \cdot)}{\alpha} \right) \right) \\
&= \arg\max_{\pi \in \prod} \mathbb{E}_a \left[ Q(s, a) - \frac{\beta}{\sqrt{|\mathcal{A}|}} \sum_{i=0}^{|\mathcal{A}|} \nabla_{a_i} Q(s, a_i) \right], \tag{22}
\end{aligned}
$$

*Proof*

$$\pi_{new}(s) = \underset{\pi \in \prod}{\arg\min} \, D_{KL}\left(\pi(\cdot|s) \Big\| \exp\left(\frac{Q(s,\cdot) + \alpha \log(\pi(\cdot|s)) - \frac{\beta}{\sqrt{|\mathcal{A}|}}\sum. \nabla. Q(s,\cdot)}{\alpha}\right)\right)$$

$$= \underset{\pi \in \prod}{\arg\min} \, \mathbb{E}_{a \sim \mathcal{A}}\left[\pi(a|s)\left(\log(\pi(a|s)) - \frac{Q(s,a) + \alpha \log(\pi(a|s)) - \frac{\beta}{\sqrt{|\mathcal{A}|}}\sum_{i=0}^{|\mathcal{A}|} \nabla_{a_i} Q(s,a_i)}{\alpha}\right)\right]$$

$$= \underset{\pi \in \prod}{\arg\min} \, \mathbb{E}_{a \sim \pi(\cdot|s)}\left[\log(\pi(a|s)) - \frac{Q(s,a) + \alpha \log(\pi(a|s)) - \frac{\beta}{\sqrt{|\mathcal{A}|}}\sum_{i=0}^{|\mathcal{A}|} \nabla_{a_i} Q(s,a_i)}{\alpha}\right]$$

$$= \underset{\pi \in \prod}{\arg\max} \, \mathbb{E}_a\left[Q(s,a) - \frac{\beta}{\sqrt{|\mathcal{A}|}}\sum_{i=0}^{|\mathcal{A}|} \nabla_{a_i} Q(s,a_i)\right], \tag{23}$$

where $D_{KL}(\cdot\|\cdot)$ is the KL divergence. □

## D  PROOF OF LEMMA 4

**Lemma 4.** *Denote $\pi_{new}$ and $\pi_{old}$ as the policies before and after the update defined in* (22), *respectively. Then the expected policy improvement, i.e., $\mathbb{E}_{(s_t,a_t)\sim\mathcal{S}\times\mathcal{A}}[Q_{\pi_{new}}(s_t,a_t) - Q_{\pi_{old}}(s_t,a_t)] \geq 0$, can be guaranteed, where $Q_{\pi_{new}}$ is the real Q-value following bellman operator and $Q_{\pi_{old}}$ is the modified value employing* (6).

*Proof*

$$\mathbb{E}_{(s_t,a_t)}[Q_{\pi_{new}}(s_t,a_t)]$$

$$= \mathbb{E}_{(s_t,a_t)}\mathbb{E}_{(s_{t+1},a_{t+1})}[r(s_t,a_t) + \gamma Q_{\pi_{new}}(s_{t+1},a_{t+1})]$$

$$\geq \mathbb{E}_{(s_t,a_t)}\mathbb{E}_{(s_{t+1},a_{t+1})}\left[r(s_t,a_t) + \gamma Q_{\pi_{new}}(s_{t+1},a_{t+1}) - \frac{\gamma\beta}{\sqrt{|\mathcal{A}|}}\sum_{i=0}^{|\mathcal{A}|}\nabla_{a_{t+1,i}} Q_{\pi_{new}}(s_{t+1},a_{t+1,i})\right]$$

$$\geq \mathbb{E}_{(s_t,a_t)}\mathbb{E}_{(s_{t+1},a_{t+1})}\left[r(s_t,a_t) + \gamma Q_{\pi_{old}}(s_{t+1},a_{t+1}) - \frac{\gamma\beta}{\sqrt{|\mathcal{A}|}}\sum_{i=0}^{|\mathcal{A}|}\nabla_{a_{t+1,i}} Q_{\pi_{old}}(s_{t+1},a_{t+1,i})\right]$$

$$= \mathbb{E}_{(s_t,a_t)}\left[Q_{\pi_{old}}(s_t,a_t) + \gamma\beta\|\nabla_{a_t} Q_{old}(s_{t+1},a_{t+1})\|_2 - \frac{\gamma\beta}{\sqrt{|\mathcal{A}|}}\sum_{i=0}^{|\mathcal{A}|}\nabla_{a_{t+1,i}} Q_{\pi_{old}}(s_{t+1},a_{t+1,i})\right]$$

$$\geq \mathbb{E}_{(s_t,a_t)}[Q_{\pi_{old}}(s_t,a_t)], \tag{24}$$

where the first equality follows the bellman operator, the first inequality holds because $\mathbb{E}_s\left[\sum_{i=0}^{|\mathcal{A}|}(\nabla_{a_i} Q_\omega(s,a_i))\right] \geq 0$, the second inequality holds because of the update rule of (22), the second equality follows the expected form of modified Bellman backup operator given by (6), and the last inequality holds because the difference of last two terms is nonnegative. □

## E  NETWORK ARCHITECTURE

We construct the critic network based on a fully-connected MLP with four hidden layers. The input is composed of the state and action, outputting a value representing the Q-value. The ReLU functions are adopted to activate the two hidden layers. To avoid the second order gradient of Q-value with respect to the action being zero, we add some nonlinearity into the fully-connected critic layers,

Table 1: **List of hyperparameters**

| Hyperparameter | Value | Description | Algorithm applied |
|:---:|:---:|:---:|:---:|
| $LR\_a$ | 0.0003 | Learning rate of actor | All |
| $LR\_c$ | 0.0003 | Learning rate of critic | All |
| $\tau\_a$ | 1 | Soft update parameter of actor | All |
| $\tau\_c$ | 0.005 | Soft update parameter of critic | All |
| $\gamma$ | 0.99 | Discount horizon factor | All |
| $var$ | 0.2 | The variance of exploration noise | All |
| $\alpha$ | 0.1 | Fixed temperature | Except DDPG and TD3 |
| $\alpha_d$ | 0.1 | Wight factor of KL regularization | BRAC |
| $\beta$ | 0.05 | newly-induced hyperparameter | GNC |
| $Batch$ | 256 | Size of each mini-batch | All |
| $Units$ | 256 | Hidden layer units | All |
| $Memory$ | 1000000 | Size of replay buffer | All |
| $Interval$ | 500 | Evaluation period | All |
| $Test$ | 10 | Rollouts per evaluation | All |

which is formulated by

$$
\begin{aligned}
L_1 &= ReLU(w_1 \cdot \left[s, a, a^3\right] + b_1), \\
L_2 &= ReLU(w_2 \cdot L_1 + b_2), \\
L_3 &= ReLU(w_3 \cdot L_2 + b_3), \\
Q &= ReLU(w_4 \cdot L_3 + b_4),
\end{aligned}
\tag{25}
$$

where $w_i$ and $b_i$ for $i \in \{1, 2, 3, 4\}$ are hidden parameters of critic layers.

The setting of policy network follows normal random distribution, whose expectation and variance are fully-connected networks fed only by the state. Both of them have two hidden layers activated by the ReLU function. After the hidden layers, a Tanh function and a Softplus function follows to form the expectation and variance, respectively. With the expectation and variance, a normal distribution can be achieved to represent the random policy. The architecture of networks are plotted in Fig. 3.

The above mentioned network architecture is adopted for the random policy. For the algorithm using the deterministic policy, the output of actor network is the expectation of normal random distribution.

## F  HYPERPARAMETERS

Table 1 lists the common hyperparameters shared by all experiments and their respective settings. In this table, $LR\_a$ means the learning rate of the actor (includes $lambda$ in our proposed algorithm), and $LR\_c$ means the learning rate of critics. $\tau\_a$ and $\tau\_c$ represent soft update hyperparameter of the actor and the critic, respectively, and $\tau\_a = 1$ means we adopt immediate update for the actor. The symbol $var$ represents the variance of gaussian exploration noise, and $\alpha$ is the fixed temperature hyperparameter, which is applied in algorithms except DDPG and TD3. $\alpha_d$ represents the weight factor of KL divergence for policy regularization applied in BRAC, and $\beta$ is fixed newly-induced hyperparameter to balance the contribution of GNC term.

Moreover, $Batch$ represents the size of mini-batches sampled for training, and $Memory$ is short for the size of replay buffer. The rest in Table 1 are the hyperparameters for the evaluation procedure, specifically, $Interval$ means how many time steps between two successive evaluation procedures, and $Test$ means the number of rollouts run during each evaluation procedure.

## G  OFFLINE BENCHMARKS AND RESULTS

In this part, we evaluate the proposed approach on the D4RL benchmark of MuJoCo tasks Todorov et al. (2012); Brockman et al. (2016); Fu et al. (2020), which includes multiple datasets related to different tasks. The benchmarks selected to evaluate the proposed offline algorithm include Ant, Humanoid, walker2d and Halfcheetah related offline tasks.

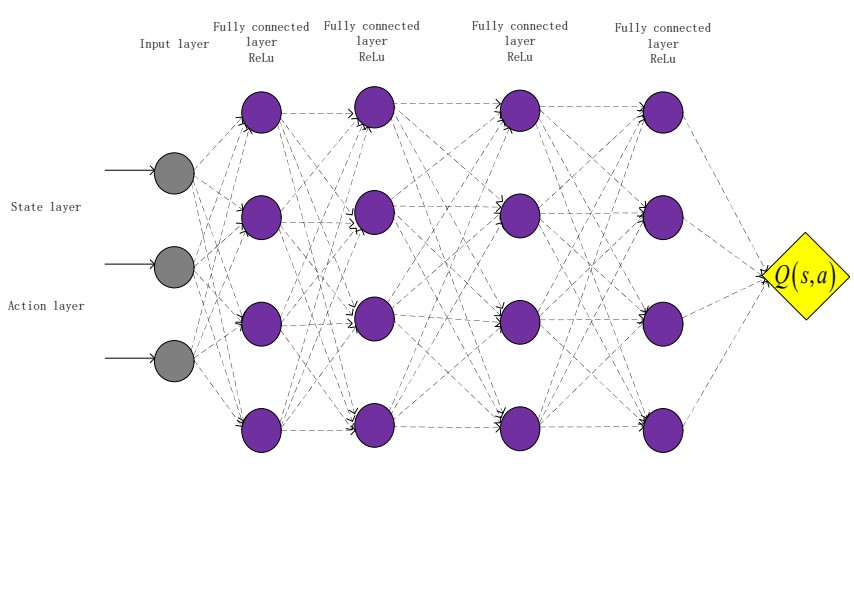

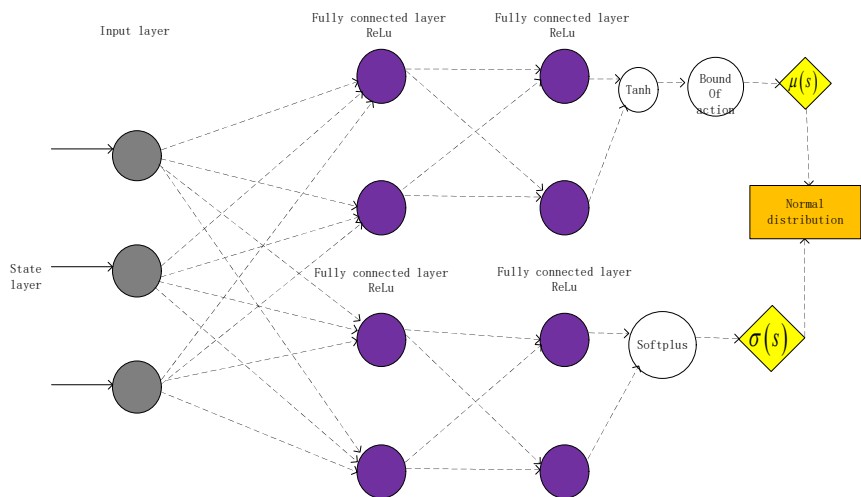

Figure 3: Architecture of networks.

The baselines we adopt to compare with our proposed algorithm are implicit q-learning (IQL) Kostrikov et al. (2021), batch-constrained deep Q-learning (BCQ) Fujimoto et al. (2019) and T-D3+BC. To ensure a fair experimental evaluation, we share the same set of hyperparameters across algorithms for the same benchmark. After some experimental trials, we determine to utilize the delayed update rule to update the target parameters $(\omega', \theta')$ instead of "soft" target updates. Specifically, we execute $\omega' \leftarrow \omega$ and $\theta' \leftarrow \theta$ every 100 steps. We set the maximum time step as $10^6$ for each algorithm and evaluate it every 5000 time steps. The evaluation process consists of 10 episodes, whose results will be averaged to get the periodical evaluation. We record the final performance results in Table and show the learning curves in Fig. 4.

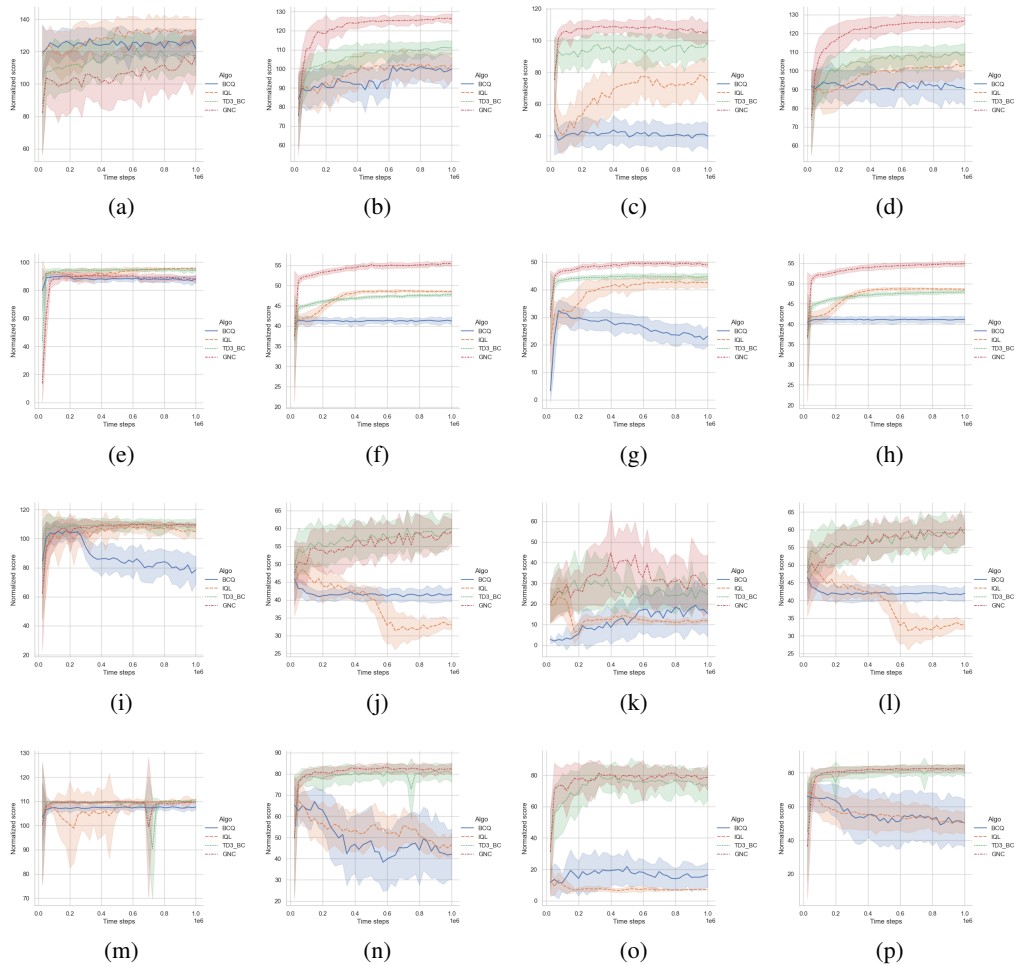

Figure 4: Normalized score versus time step in (a) Ant-expert; (b) Ant-medium-expert; (c) Ant-medium-replay; (d)Ant-medium; (e) Halfcheetah-expert; f Halfcheetah-medium-expert; (g) Halfcheetah-medium-replay; (h) Halfcheetah-medium; (i) Hopper-expert; j Hopper-medium-expert; (k) Hopper-medium-replay; (l) Hopper-medium; (m) Walker2d-expert; (n) Walker2d-medium-expert; (o) Walker2d-medium-replay; (p) Walker2d-medium

From these figures, we can observe that the performance of GNC-BC (labeled as "GNC" in these plots) is close to that of TD3+BC when the dataset is expert, because expert demonstrations make the behavior cloning reliable. In other word, good dataset reduces the necessity of policy exploration and thus enhances the dependence on behavior cloning. Another observation is that GNC-BC has much higher converged value and better stability than other baselines except for the Hopper task, considering the benefits of GNC policy exploration to skip suboptimal points. As for the Hopper task, it is easy to gain policy improvement without guarantee of stability, according to our empirical trials.

