# OpenReview forum: "Gradient-norm Constrained Algorithm on Offline and Online Learning"
_ICLR.cc/2024/Conference — ICLR 2024 Conference Withdrawn Submission_

### Official Review · Reviewer_daqm · 2023-10-19

**Soundness:** 1 poor
**Presentation:** 2 fair
**Contribution:** 1 poor
**Rating:** 1
**Confidence:** 5

**Summary:**

The paper proposes a regularization method for training value and policy networks in offline & online RL settings. First, the paper incorporates a gradient-norm penalty to the target value that encourage the policy network to avoid choosing actions with high sharpness in the Q-function. Second, the paper introduces a regularizer based of the Q-function gradient that encourages the policy network to avoid the stationary points in the parameter space. The paper claims that the proposed method outperforms previous method in both offline and online settings.

**Strengths:**

- Introducing a Q-function gradient norm regularizer to penalize overestimate Q-values is novel.

**Weaknesses:**

- The proof of Proposition 1 within Appendix B appears to be wrong and necessitates a thorough review. Specifically, there is ambiguity regarding the representation of "$a$", which requires clarification: is it to be interpreted as a distribution or a vector? Also, since $\pi(a \mid s)$ is multivariate, its inverse cannot be defined. Moreover, the third line in Equation 21 seems wrong since the left matrix is of dimension $1 \times |\mathcal{A}|$ but the right matrix is of dimension $1 \times \theta$, which cannot be multiplied.
- The paper omits several important baselines in both online and offline settings, including REDQ [1], EDAC [2], and PBRL [3].
- The performance improvement in the offline setting seems very marginal.

[1] Xinyue Chen et al., Randomized Ensembled Double Q-Learning: Learning Fast Without a Model, ICLR 2021\
[2] Gaon An et al., Uncertainty-Based Offline Reinforcement Learning with Diversified Q-Ensemble, NeurIPS 2021\
[3] Chenjia Bai et al., Pessimistic Bootstrapping for Uncertainty-Driven Offline Reinforcement Learning, ICLR 2022

**Questions:**

See the weaknesses above.

---

### Official Review · Reviewer_FbbA · 2023-11-01

**Soundness:** 3 good
**Presentation:** 3 good
**Contribution:** 2 fair
**Rating:** 5
**Confidence:** 3

**Summary:**

The paper introduces gradient norm regularization for off-policy actor-critic methods for deep reinforcement learning, addressing key issues like the saddle point problem during parameter updates and distribution mismatches in off-policy methods. Additionally, the method is combined with behavior cloning for offline reinforcement learning applications.

**Strengths:**

1. The paper targets significant issues in RL, such as the saddle point problem and distribution mismatch in off-policy methods.
2. The proposed policy regularization and value penalty provide some potentially new perspectives to address these challenges with theoretical intuitions.
3. Empirical results show some benefits of the proposed regularization methods.

**Weaknesses:**

1. In the experimental section, only DDPG, TD3, and SAC are included, and no other regularization-based approaches are included.  As introduced in the related work, there are many other approaches with regularizers and those could be missing baselines.
2. Introducing gradient norm regularization would result in extra computation overhead since it needs to calculate the second-order derivative, and there is no discussion on the comparison of the compute. See Q2.
3. There is no ablation of how sensitive the proposed method is with respect to the extra hyperparameter \beta.

**Questions:**

Q1. How is the sum over action space and the size of the action space in eq. (15) calculated in the continuous domain? I could imagine some discretization tricks but there is no reference for that in the paper.

Q2. There could be some trade-off between computation overhead and performance. I wonder whether other simpler regularization methods, like the ones mentioned in the related works without second-order derivatives, could achieve similar performances.

---

### Official Review · Reviewer_ar2S · 2023-11-02

**Soundness:** 2 fair
**Presentation:** 1 poor
**Contribution:** 2 fair
**Rating:** 3
**Confidence:** 3

**Summary:**

This paper identifies two pertinent problems with off-policy RL algorithms, namely: the saddle point problem and value function exploitation problem. The authors propose to tackle them with 2 regularization objects around the Q-value gradients.

To mitigate the value function exploitation problem, the authors introduce a penalty term in the value target when performing policy iteration. Specifically, the penalty is a scaled norm of the Q-value's gradient with respect to actions. The hypothesis is that sharp Q-value landscapes could push the policy into OOD regions. To tackle the saddle point problem, the authors add an extra term when computing policy updates to encourage actions with non-negative gradients in the Q-value function. The rationale behind this is a pessimistic view where the action produced by the current policy is always treated as a saddle point.

The authors experiment with the proposed method in both online and offline RL settings on a few Mujoco locomotion tasks. The results show that the policies trained by the proposed GNC method generally converge to higher performance.

**Strengths:**

- This work studies two fundamental challenges with off-policy RL with function approximators. A working solution has the potential for high impact.
- The authors provide mathematic derivations to justify the penalty terms.
- The experiment results support that GNC is outperforming SOTA RL algorithms.

**Weaknesses:**

The main weakness of this paper is in the way it is presented.
- Typos, wrong references, and dubious claims: There is a typo in the first sentence of the paper, which is not a good look. References to equations (10 vs. 22) (proposition 1 vs. 2) are intertwined, making it hard to read. Finally, the paper says RL is mainly limited to video games but in reality, RL has been applied to robot learning, real robot hardware, and even the training of large language models.
- Poorly organized figures: In Figure 2, the text is not legible. The ordering of the experiments is inconsistent between the main results and the ablation studies. The plots of the ablation study should also contain the base GNC method.
- The offline RL setting is potentially a good selling point of this paper, but the results are in the appendix.
- The derivation of policy regularization might need some clarification (see questions).

**Questions:**

- GNC-VP seems to be completely identical to GNC in the humanoid environment. Maybe I missed the experiment details, but what would be the cause?
- I don't fully understand the derivation of the objective for policy regularization. In equation (8), the expected gradient w.r.t. policy parameters $\theta$ is the same shape as $\theta$. Is it greater than or equal to zero element-wise? Also, the first paragraph of section 4.2 states that the gradient of Q w.r.t. actions should be non-negative, but in equation (10), the update rule seems to be penalizing positive gradients. Am I missing something here?